# Improved UAV Opium Poppy Detection Using an Updated YOLOv3 Model

**DOI:** 10.3390/s19224851

**Published:** 2019-11-07

**Authors:** Jun Zhou, Yichen Tian, Chao Yuan, Kai Yin, Guang Yang, Meiping Wen

**Affiliations:** 1University of Chinese Academy of Sciences, Beijing 100049, China; zhoujun17@mails.ucas.edu.cn; 2Aerospace Information Research Institute, Chinese Academy of Sciences, Beijing 100101, China; yinkai@radi.ac.cn (K.Y.); yangguang@radi.ac.cn (G.Y.); wenmp@radi.ac.cn (M.W.)

**Keywords:** UAV, opium poppy, object detection, YOLOv3 model, deep learning, CNN, spatial pyramid pooling, GIoU

## Abstract

Rapid detection of illicit opium poppy plants using UAV (unmanned aerial vehicle) imagery has become an important means to prevent and combat crimes related to drug cultivation. However, current methods rely on time-consuming visual image interpretation. Here, the You Only Look Once version 3 (YOLOv3) network structure was used to assess the influence that different backbone networks have on the average precision and detection speed of an UAV-derived dataset of poppy imagery, with MobileNetv2 (MN) selected as the most suitable backbone network. A Spatial Pyramid Pooling (SPP) unit was introduced and Generalized Intersection over Union (GIoU) was used to calculate the coordinate loss. The resulting SPP-GIoU-YOLOv3-MN model improved the average precision by 1.62% (from 94.75% to 96.37%) without decreasing speed and achieved an average precision of 96.37%, with a detection speed of 29 FPS using an RTX 2080Ti platform. The sliding window method was used for detection in complete UAV images, which took approximately 2.2 sec/image, approximately 10× faster than visual interpretation. The proposed technique significantly improved the efficiency of poppy detection in UAV images while also maintaining a high detection accuracy. The proposed method is thus suitable for the rapid detection of illicit opium poppy cultivation in residential areas and farmland where UAVs with ordinary visible light cameras can be operated at low altitudes (relative height < 200 m).

## 1. Introduction

Illegal drugs can degrade physical and mental health while affecting social stability and economic development. The rapid detection of illicit opium-poppy plants is integral to combatting crimes related to drug-cultivation. Satellite remote sensing has traditionally played an important role in monitoring poppy cultivation. Taylor et al. [1], along with the U.S. government, used satellite remote sensing to detect poppy plots in Afghanistan for several years. Liu et al. [2] used ZY-3 satellite imagery to detect poppy plots in Phongsali Province, Laos, using the single-shot detector (SSD)-based object detection method. Jia et al. [3] studied the spectral characteristics of three different poppy growth stages, showing that the best period for distinguishing poppy from coexisting crops was during flowering. However, new cultivation strategies such as planting small, sporadic, or mixed plots make it more difficult to identify small-scale cultivation in non-traditional settings, such as courtyards. Compared to satellite remote sensing, unmanned aerial vehicles (UAVs) capture images with much higher spatial resolution (<1 cm). UAV platforms are highly flexible: they are able to conduct observations under broader conditions and can fly closer to the ground to capture finer textural features. This ability to capture such detailed features together with the lower cost of UAVs compared to satellite remote sensing are rapidly making UAV systems both an effective alternative and a supplement to satellite remote sensing, particularly in the detection of illegal poppy cultivation.

Poppy identification in UAV images is currently conducted primarily via visual interpretation because of major differences in the characteristics of different growing stages and the complexity of planting environments. A skilled expert usually requires at least 20 s to detect poppy via visual interpretation of a UAV image. This requires extensive human and material resources given the sheer quantity of UAV images that can be collected. Machine learning methods based on manual design features perform well only under limited conditions; such limited conditions currently do not sufficiently account for variation in altitude, exposure, and rotation angle, all of which can significantly affect the appearance of similar ground objects in UAV images and add difficulty to feature recognition. Therefore, a new method to improve work efficiency and detection accuracy is urgently needed; for this purpose, the ongoing development of deep-learning-based object detection holds great promise.

Deep learning has rapidly developed since its initial proposal in 2006, and especially after 2012. Techniques represented by deep convolutional neural networks (DCNNs) have been widely used in various fields of computer vision, including image classification [4,5,6,7], object detection [8,9,10,11,12,13], and semantic segmentation [14,15,16,17,18]. Compared with traditional machine learning methods based on manual design features, DCNNs have a more complex structure that is capable of extracting deeper semantic features and learning more powerful general image representations. Currently, convolutional neural networks (CNNs) are mainly composed of several convolution layers that may include pooling layers, followed by several full connection layers. The feature map generated by the convolution layer is usually activated by the rectified linear unit (ReLU) and regularized by batch normalization (BN) [19] to prevent network overfitting. Researchers have continuously expanded network depth and width or reduced the complexity of the network model to improve the accuracy or speed of image classification; alongside such advances, complex networks have been proposed, such as the Inception series [5,19,20,21] and Residual series models [7,22], which expand the width and deepen the network layer, or the lightweight networks, such as SqueezeNet [23], MobileNet [24,25,26], and ShuffleNet [27,28].

CNN’s successful performances in image classification tasks has advanced the development of object detection. Traditional object detection relies on a search framework based on sliding windows, which divide a graph into several sub-graphs with different positions and scales; a classifier is then used to distinguish parts that do not contain specified objects by sub-graph. This method requires designing different feature extraction methods and classification algorithms for different objects. Object detection methods based on deep learning are mainly divided into two categories based on region proposal and regression. Region proposal methods (such as Regions with CNN features (R-CNN) [8], Fast R-CNN [29], and Faster R-CNN [9]) mainly use texture, edge, color, or other information in the image to determine the possible location of an object in the image in advance and then use the CNNs to classify and extract the features of these locations. Although this method can achieve good accuracy, it is difficult to implement in real-time detection. Regression methods (such as OverFeat [30], You Only Look Once (YOLO) [10,31,32], and SSD [11,33]) use a single end-to-end CNN to directly predict the location and category of an object’s bounding box in multiple locations within the image, greatly accelerating the speed of object detection.

Deep-learning-based object detection methods have been widely used in remote sensing applications. For example, Ammour et al. [34] combined the CNN and support vector machine (SVM) methods to conduct vehicle identification research using aerial photographs. Bazi and Melgani [35] constructed a convolutional support vector machine network (CSVM) for the detection of vehicles and solar panels using an UAV dataset. Chen et al. [36] used Faster R-CNN object detection to identify airports from aerial photography. Rahnemoonfar et al. [37] built an end-to-end network (DisCountNet) to count animals in UAV images. Ampatzidis and Partel [38] used the YOLOv3 model with normalized difference vegetation index (NDVI) data to detect trees in low-altitude UAV photos.

YOLOv3 is one of the state-of-the-art one-stage detection networks; the detection speed is very fast and detection accuracy is quite high in the current one-stage detection model. The YOLOv3 model has been successfully applied in the field of remote sensing and UAV. Given these successful past applications, we chose to base this study on the YOLOv3 model. Firstly, we used the beta distribution as a ratio to mix-up backgrounds and objects, then applied random augmentation metrics to enlarge the dataset. Secondly, we assessed the performance of various backbone networks, added a spatial pyramid pooling unit, and used the generalized intersection over union (GIoU) method to compute the bounding box regression loss. Thirdly, we used various evaluation metrics to assess the model results in terms of superiority, efficiency, and model applicability. The remaining sections of the paper are presented in the following order: study area and data, research methods, model evaluation metrics, results, discussion, and conclusions.

## 2. Study Area and Data

### 2.1. Data Acquisition

In most parts of mainland China, the best growing season for opium poppy is March–August. Due to scale effects, opium poppy tokens under different flying heights show entirely different characteristics. Thus, we selected UAV images collected from 2014 to 2018 that were verified to contain poppies by K.Y., G.Y., and M.W. All photos were taken within March–August, every year from 2014 to 2018, using two UAV styles at different relative heights: (1) a DJI UAV (camera sensor of 13.2 × 8.8 mm^2^, focal length of 8.8 mm and a photo size 5472 × 3648 pixels) took images at altitudes of 30 and 60 m; (2) a fixed-wing UAV (SONY A7R2 camera, camera sensor of 36 × 24 mm^2^, lens focal length of 35 mm, photo size of 7952 × 5304 pixels) took images at an altitude of 150 m.

The ground resolution of the images (ground sampling distance of the image, i.e., GSD) could be calculated as:(1)GSD=H ×a / f,
where *f* is the focal length of the photographic lens, and *a* is the pixel size; *a* can be calculated as:(2)$$a=\sqrt{\frac{S_{pe}}{S_{p}}}=\frac{L_{pe}}{L_{p}},$$
where $S_{pe}$ is the photosensitive element size, $S_{p}$ is the photo size, $L_{pe}$ is the length of photosensitive element size, and $L_{p}$ is the length of photo.

At the relative flying height of approximately 30, 60, and 150 m, the ground resolution of the images was approximately 0.8, 1.6, and 2.0 cm, respectively. In most parts of China, the poppy seedling stage occurs before April, the flowering period ranges from April to June, and the fruiting period ranges from July to August. The opium poppy leaves in the seedling stage are grayish green while the flowering stage is characterized by the presence of symbiotic plants [3], where the poppy seeds are long ellipsoids. Poppy monitoring is mainly performed from March to August, monitoring the poppy in the seedling stage and flowering period, whereas data in the fruit period is negligible. Here, poppy photos that involved seedling and flowering were selected. Figure 1 shows the characteristics of poppies at different growing periods and flying heights. As the diameter of a single opium poppy is approximately 30 cm, poppy textures can be clearly observed from low altitudes (i.e., a relative height < 50 m). At a relative height of 50–100 m, individual poppies and smaller features can still be distinguished; however, at >100 m, flowering poppies can only be identified based on observer experience, and seedling poppies are typically even more difficult to identify. All severe overexposures and blurred photos were removed from the image dataset, leaving 1040 photos selected for this study: 495 photos were taken at 30 m, 395 at 60 m, and 150 at 150 m.

### 2.2. Data Processing

#### 2.2.1. Preliminary Processing

Only a small area of any given image contained poppies. According to the ground-authenticated image, the poppy was accurately marked in the original image (Figure 2). The labelImg tool [39] was used to mark and generate corresponding labeled information, which was then randomly cut according to the location of poppies, reducing it to 416 × 416 pixels. The specific implementation is shown in Algorithm 1.

**Algorithm 1:** Data Cropping Strategy**Input**: One dataset, *A*, including *N* big UAV images.**Output**: One dataset, *B*, including cropped images with a fixed size (416 × 416 pixels).1:*B*←{}2:**for***a***in***A*:3:*objs*←all objects in *a*4:**for***obj***in***objs*:5:*objs*←*objs*\{*obj*}6:*b*←random crop image, *a*, to fixed size according to the bounding box of *obj*7:*B*←*B*∪{*b*}8:**for***o***in***objs*:9:**if** the intersection over union the between *b* and *o* is bigger than 0.5:10:*objs*←*objs*\{*o*}11:**end if**12:**end for**13:**end for**14:**end for**15:**return***B*

Flying height and growth period strongly affected the quantities of images containing poppies. More images containing poppies were found at 30 m than at 150 m and during the flowering stage rather than the seedling stage (Table 1). To balance the number of poppy samples at different heights and growth periods, a random replication (oversampling) was used to replicate the data for small samples.

#### 2.2.2. Data Fusion

As the poppy cultivation environment is complex and positive sample data are limited, it is difficult for labeled poppy samples to truly reflect the majority of the cultivation environment. Zhang et al. [40] successfully applied the mix-up method to image classification to enhance the generalization ability. Zhang et al. [41] studied the natural co-occurrence of objects that played an important role in object detection and used Beta (1.5, 1.5) as a ratio to mix-up two pictures of different objects to obtain a high recall rate. This approach has been mainly used to enhance the generalization performance of image classification and object detection by merging the training images with different objects by pixel-by-pixel mixing. Here, 1000 photos without poppies were randomly cut from the original UAV photos to be used as background. The mixup method based on a Beta distribution was then used to fuse the background and positive samples to form new samples.

The probability density function of the Beta distribution is shown in Figure 3. For the Beta distribution with parameters alpha and beta, i.e., Beta (alpha, beta), when alpha and beta are both 0.2, the Beta distribution concentrates near 0 or 1, which is usually used as a mix-up ratio in image classification. When alpha and beta are both 1.0, the Beta distribution is uniform. When alpha and beta are both greater than 1, the Beta distribution is concentrated near 0.5, which is used as a mix-up ratio in object detection to mix up two objects from different training images. Beta (0.2, 0.2) enables the image to introduce some background information while maintaining most of the information of the positive sample of a real poppy image. Thus, in this study, we selected Beta (0.2, 0.2) to mix up background and ground truth data; the mixed image was calculated as:(3)$$P_{new}=\{\begin{array}{c} \mu \times P_{bg}+\left(1-\mu \right)\times P_{gt},\;\;if\;\mu <0.5\\ \left(1-\mu \right)\times P_{bg}+\mu \times P_{gt},\;\;if\;\mu \geq 0.5 \end{array},$$
where $P_{new}$ is the fused image, $P_{bg}$ is the background image, $P_{gt}$ is the real poppy image, and μ is a coefficient that conforms to Beta (0.2, 0.2) and is between 0 and 1. A fused image is shown in Figure 4.

#### 2.2.3. Data Augmentation

The balanced data set contained 1235 photos. To further expand the size of the dataset, random data augmentation was performed on the existing data. This involved, first, random position transformation including random cropping (with limitation), flipping (including random horizontal and vertical flipping), rotation, and resizing (with random interpolation); second, random color adjustment, including random changes in the brightness, contrast, sharpening, and noise addition (including salt and pepper and Gaussian noise).

Each data augmentation operation included 2–4 instances of random position transformation or random color adjustment followed by resizing to 416 × 416 pixels, after which all data were combined to form the dataset used in this study. The detailed procedure is described in Algorithm 2. Seventy percent of all samples were randomly selected as the training dataset, 10% as the validation dataset, and the remaining 20% as the testing samples (Table 2).

**Algorithm 2**: Data Augmentation Strategy**Input**: The original dataset, *A*, with *N* images and a random transform method set, *T* (including random cropping, random flipping, random rotation, random resizing, random changes in brightness, random sharpening operation, and random noise addition).**Output**: Enhanced dataset *B*.1:*B*←*A*2:**for***a***in***A***do**:3:*aug_num* = random (2, 4)4:**for***i*←0 to *aug_num***do**:5:*b*←*a*6:*times*←07:**while***times* < 2 **do**:8:randomly select a transform method from set *T*, *b*←transform image *b*9:*times*←*times*+110:**end while**11:resize *b* to 416 × 416 pixels12:*B*←*B*∪{*b*}13:**end for**14:**end for**15:**return***B*

## 3. Methodology

### 3.1. YOLOv3 Model Based on Multiple Backbone Networks

As the feature extractor in object detection networks, the backbone network plays an important role in object detection. To a large extent, backbone networks determine the speed and accuracy of the detection network. Complex backbone networks will significantly improve the detection accuracy but will also seriously affect the detection speed, whereas lightweight networks have the opposite effect. Three complex networks (DarkNet53, ResNet50, and DenseNet121) and two lightweight networks (MobileNetv2 and ShuffleNetv2) were tested.

#### 3.1.1. Backbone Networks

The YOLO series of object detection networks used DarkNet as the backbone network; similarly, the DarkNet53 network was the basic network used in YOLOv3. This network consists of several consecutive 1 × 1 and 3 × 3 convolutions, introducing a residual structure. The network is more powerful than YOLOv2′s DarkNet19 network and more efficient than Inceptionv3 and ResNet101. A brief review of other backbone networks tested herein is given as follows:

ResNet [7] borrows an idea from Highway Networks [42] and proposes a shortcut connection structure that allows the network to directly skip one or two layers to form residual units (Figure 5a). ResNet is an excellent image classification network and is widely used in semantic segmentation and object detection.

DenseNet [43] adopts a dense connection structure in which the dense block connects all layers together (Figure 5b). The dense block structure greatly reduces the number of network parameters and, to a certain extent, alleviates the problem of gradient disappearance and model degradation.

MobileNetv2 [25] is a state-of-the-art lightweight network that adopts the inverted residual structure (Figure 5c). The structure reduces the number of parameters and complexity of the network model and accelerates the forward propagation of the network.

ShuffleNetv2 [28] adopts channel decomposition and channel shuffling methods (Figure 5d), greatly increasing speed while maintaining high precision.

#### 3.1.2. Model Training

To compare the accuracy and efficiency of different models, we adopted unified hyper-parameters for training the different backbone networks (Table 3). We set beta_1 = 0.9, beta_2 = 0.999, and weight_decay = 0.0001 as the parameters of the optimization method reported in Adam [44] to optimize the network. Since YOLOv3 (based on DarkNet53) provided the official weights, there were no official pre-trained weights for the other four backbone networks during initialization, thus the random initialization method was used to uniformly initialize the weights. Likewise, for YOLOv3, based on DarkNet53, the official weights were not used.

### 3.2. Improved YOLOv3 Model

#### 3.2.1. Improved Spatial Pyramid Pooling Unit

Multiscale prediction in YOLOv3 connects the global features of multiscale layers for three different prediction stages but neglects multiscale local features. We developed an improved spatial pyramid pooling (SPP) unit for use with YOLOv3, using this to extract the multiscale global features of different stages and multiscale local features of the same prediction stage.

SPP was first proposed in 2015 [45]. The original SPP structure model (Figure 6) divided each feature map into a number of different grid sizes (such as 4 × 4, 2 × 2, and 1 × 1) and then performed maximum pooling operations for each grid. This resulted in C layer feature maps forming 16 × C, 4 × C, or 1 × C dimensional feature maps; these three feature maps were then finally concatenated to form a fixed-length feature map that connected into the back of the fully connected layer.

The improved SPP unit used herein pools input feature maps with different sizes, in which all of the strides are 1 and padding operations are used to maintain a fixed shape (Figure 7). The original input feature maps are then connected with all pooling results to form a SPP unit in which the filter size of the pooling layer $\left(S_{pool}\right)$ is calculated as:(4)$$S_{pool}=\;\lceil \frac{S_{map}}{n}\rceil ,$$
where $S_{map}$ is the feature map size of the input layer, such that for *n* = 1, 2, 3, three different sizes of filters are produced, respectively: $\lceil S_{map}/1\rceil \times \lceil S_{map}/1\rceil ,$
$\lceil S_{map}/2\rceil \times \lceil S_{map}/2\rceil ,$ and $\left\lceil S_{map}/3\rceil \times \lceil S_{map}/3\right\rceil$
(⌈a⌉ represents the smallest integer not less than *a*). In the experiments, there were three predictions with different feature map sizes (13 × 13, 26 × 26, and 52 × 52), allowing nine different filter sizes: 13 × 13, 7 × 7, and 5 × 5; 26 × 26, 13 × 13, and 9 × 9; and 52 × 52, 26 × 26, and 18 × 18. In the second and third prediction stages, the filter sizes were close to the previous stage. Therefore, only one SPP unit was selected for the first stage of the study (Figure 8).

The improved SPP unit used herein differs from the SPP net proposed by He et al. [45]; in the SPP net by He et al., the feature maps are divided into several grids of different sizes and max-pooling is used to pool the grid to form feature maps of different sizes. The approach used herein only uses different filter sizes for the feature maps to be pooled and uses padding to maintain unchanged dimensions.

#### 3.2.2. Network Hyperparameter Setting and Model Training

Similar to YOLOv3, each bounding box in the network predicts *bx*, *by*, *bw*, *bh,* and *confidence*, in which (*bx*, *by*) refers to the center coordinates of the prediction bounding box, (*bw*, *bh*) refers to the width and height of the prediction bounding box, respectively, and *confidence* refers to the intersection over the union between the prediction bounding box (*bx*, *by*, *bw*, *bh*) and any ground truth (*gx*, *gy*, *gw*, *gh*). Additionally, each grid unit in the network predicts the conditional probability of each category.

Here we propose a new loss function consisting of three parts: coordinate regression loss, confidence loss, and classification loss. Confidence loss and classification loss are defined as in YOLOv3; coordinate regression loss is described by *GIoU* [46] and calculated as follows:(5)$$IoU=\;\frac{B\;\cap \;G}{B\;\cup \;G},$$
(6)$$\;GIoU=IoU-\frac{C\;\backslash \;(B\cup G)}{C},\;$$
where *C* is the smallest closed convex object containing the prediction box and ground truths, *B* refers to the predicted bounding boxes, and *G* refers to the ground truths. Based on these equations, when the overlap between *B* and *G* is large, both *GIoU* and *IoU* are near 1 (only when *B*
 ∩*G* = *B*
 ∪*G* does *IoU* = *GIoU* = 1). If there is no overlap between *B* and *G*, *IoU* is near 0 while *GIoU* is less than 0 and gradually approaches –1, as the distance between *B* and *G* increases. Therefore, the range of *IoU* is [0, 1] whereas the range of *GIoU* is (–1, 1]. The bounding box regression loss can then be calculated using *GIoU*:(7)$$Coord_{loss}=\;1-GIoU.$$

As the range of values for *GIoU* is (–1, 1], the range for $Coord_{loss}$ is [0, 2). Larger values result in larger distances between the prediction box and ground truth. Relative to the mean square error (MSE) of the regression loss in the center coordinates and the width and height of the bounding box adopted in YOLOv3, the coordinate regression loss based on *GIoU* is independent of the shape and size of the bounding box and can more accurately reflect the distance between the prediction box and ground truths. The confidence loss and classification loss can be calculated as follows:(8)$$Conf_{loss}=\overset{s^{2}}{\underset{i=0}{\sum }}\overset{B}{\underset{j=0}{\sum }}1_{ij}^{obj}\left[{\left(C_{i}-{\hat{C}}_{i}\right)}^{2}\right]\;+\;\lambda_{noobj}\overset{s^{2}}{\underset{i=0}{\sum }}\overset{B}{\underset{j=0}{\sum }}1_{ij}^{noobj}\left[{\left(C_{i}-{\hat{C}}_{i}\right)}^{2}\right],$$
(9)$$\;Class_{loss}=\overset{s^{2}}{\underset{i=0}{\sum }}1_{ij}^{obj}\underset{c\epsilon classes}{\sum }{\left(p_{i}\left(c\right)-{\hat{p}}_{i}\left(c\right)\right)}^{2},\;$$
where $1_{i}^{obj}$ refers to whether the object is in grid cell *i* and $1_{ij}^{obj}$ indicates that the prediction is determined by the *j-th* bounding box predictor in grid cell *i.* The loss function is thus defined as:(10)$$Loss=Coord_{loss}+Conf_{loss}+Class_{loss}.$$

Table 4 lists the other network hyperparameters. When using *GIoU* as the loss function, training is difficult and prone to the vanishing gradient phenomenon. Therefore, we selected trained weights that did not use *GIoU* to initialize the weights of the network. We selected beta_1 = 0.9, beta_2 = 0.999, and weight_decay = 0.0001 as the parameters for the optimization method reported in Adam [44] to optimize the network.

### 3.3. Trained Model Prediction

#### 3.3.1. Single UAV Image Prediction

A single UAV image only requires direct prediction. As UAV photos are usually much larger than the required 416 × 416 pixels, direct resizing required by input will significantly reduce the image quality and detection accuracy. Thus, we used the sliding window method with a window and step size of 416 × 416 pixels to ensure no overlap between adjacent windows. This accelerated the detection speed and avoided a large number of redundant detection results.

#### 3.3.2. Multiple UAV Image Prediction

For multiple UAV images, we adopted two prediction methods. For large-scale images with low overlap, multiple images were regarded as numerous single images. The sliding window method was then used to predict each one and output the predicted results. For UAV images with high overlap, the single sheet prediction method produces a large amount of redundancy and seriously affects the detection speed. Thus, we spliced and ortho-rectified all images in ArcGIS pro 2.3 and then predicted the spliced images using sliding windows.

## 4. Model Evaluation Metrics

### 4.1. Intersection over Union

Intersection over Union (IoU) refers to the overlap ratio between two bounding boxes, calculated as:(11)$$IoU=\frac{Area\;of\;Overlap}{Area\;of\;Union}\;=\;\frac{GT\;\cap DR}{GT\;\cup DR},$$
where *GT* refers to the ground truth of the samples and *DR* refers to the detection results of the samples. By setting an appropriate overlapping threshold, the detector determines whether the box is classified as background or as a specified category (this study used only one classification, i.e., poppy). If the IoU is greater than the threshold, the box is classified as poppy. If the IoU is lower, it is classified as background.

### 4.2. Precision ×Recall Curve and Average Precision

Precision and recall are often used to evaluate the quality of a model. Precision refers to the proportion of correctly detected objects in all detected objects whereas recall refers to the proportion of correctly detected objects in all positive samples detected. True Positive (*TP*, i.e., a correct detection with an IoU of no less than the threshold), False Positive (*FP*, i.e., a wrong detection with an IoU of less than threshold), False Negative (*FN*, i.e., a ground truth not detected), and True Negative (*TN*) are usually used to calculate recall and precision as follows:(12)$$Recall=\;\frac{TP}{TP+FN},$$
(13)$$Precision=\;\frac{TP}{TP+FP}.$$

The precision × recall (PR) curve is a good method to evaluate the performance of object detectors. This method draws a curve for each class according to the change in confidence. An object detector is considered good if its precision remains high as the recall rate increases, indicating that the precision and recall rate can still remain high if there is a change in the confidence threshold. Such a curve was drawn using the precision and recall rate values. The area under the PR curve represented the average precision (*AP*):(14)$$AP=\;\int_{0}^{1}P\left(R\right)dR,$$
where *P* refers to the precision and *R* refers to the recall rate. The threshold value of the IoU was set to 0.5 and the AP named as *AP50*, which was used to evaluate the model.

### 4.3. Mean Average Precision

Each class *i* has a corresponding AP $\left(AP_{i}\right)$, where the mean AP (*mAP*) refers to the mean of the AP for each class:(15)$$mAP=\;\frac{\sum_{i=1}^{n}AP_{i}}{n},$$
where *n* represents the number of all categories to be predicted. Here, the *mAP* was equal to the *AP* because there was only one category (poppy).

### 4.4. F-Score

When using precision and recall rate evaluation indices, high indices are ideal; however, in general, it is difficult to simultaneously achieve high precision and recall rates. Therefore, a trade-off between the precision and recall rate according to the actual situation is necessary. The *F*-score was thus introduced to comprehensively consider the harmonic value of the precision and recall rate:(16)$$F=\;\frac{\left(1+\beta^{2}\right)*P*R}{\beta^{2}*P+R},$$
where β refers to the harmonic coefficient between *P* and *R*.

The *F*-score is the harmonic average of the precision and recall rate. When *beta* is greater than 1, the recall rate is more important; when *beta* is less than 1, the precision is more important. In actual poppy detection, more attention should be paid to the recall rate. Therefore, a *beta* value of 2 was selected to obtain:(17)$$F2=\;\frac{5*P*R}{4*P+R}.$$

Therefore, the *F2* score value ranges from 0 to 1.0, indicating that when *Precision* = *Recall* = 1, the *F2* score reached a maximum of 1.0.

## 5. Results

The detectors were based on the Keras 2.24 framework with a Tensorflow 1.12 backend. All experiments were conducted using a server with the following characteristics: CPU: Intel Core I9-9900K, GPU: NVIDIA RTX 2080Ti, Memory: 32 GB, Hard drive: Intel 660P SSD (QLC flash granule) 512 GB.

### 5.1. Backbone Network Assessment

#### 5.1.1. Training

In this stage, the convergence speed and decline of the training and validation loss were compared for each of the different backbone networks (Table 5). The YOLOv3 model took approximately 257 epochs to converge using DarkNet53, 251 using DenseNet121, 374 using ResNet50, 346 using MobileNetv2, and 276 using ShuffleNetv2. DenseNet121 yielded the fastest convergence performance. Several outliers were directly eliminated and filled with the average of the adjacent point; the training and validation losses for all backbone networks were then plotted (Figure 9), showing that ResNet50 had the smallest training loss whereas DenseNet121 had perfect validation loss.

#### 5.1.2. Testing

For the different backbone networks, the AP, detection speed, and *F2* -score were compared. The PR curves (Figure 10) and *F2* score-recall curves (Figure 11) show that DenseNet121 and ResNet50 produced PR curves more inclined to the upper right corner, with larger areas underneath and perfect *F2* scores. DarkNet53 and ShuffleNetv2 produced PR curves inclined toward the bottom right corner, with smaller areas underneath and imperfect *F2* scores. Unexpectedly, the PR curve for MobileNetv2 fell between that for DenseNet121 and DarkNet53, and the area under its PR curve was bigger than that for DarkNet53.

Table 6 lists the AP, model parameters, detection time in the testing dataset, and *F2* score for all backbone networks. Similar to the PR curve results, ResNet50 had the highest AP (95.60%) and the best *F2* score (0.956) but also had the largest model parameters (419.6 MB), longest detection time (38.9 s), and slowest speed (21.9 FPS). Contrastingly, ShuffleNetv2 had the fastest detection speed (33.3 FPS) and smallest model parameters (80.1 MB) but a lower AP (91.09%) and *F2* score (0.913). DenseNet121, DarkNet53, and MobileNetv2 had moderate performances with APs, parameters, detection speeds, and *F2* scores between those of ShuffleNetv2 and ResNet50.

MobileNetv2 had an AP of nearly 95% (only 0.85% lower than that of ResNet50) but was faster (29.2 FPS), with model parameters of 136.3 MB (32% of ResNet50) and an *F2* score of 0.942 (0.011 lower than that of ResNet50 but 0.029 higher than that of ShuffleNetv2). Overall, MobileNetv2 provided the most balanced model with the best trade-offs for accuracy and speed. Therefore, MobileNetv2 was selected as the backbone network for the YOLOv3 model in subsequent experiments.

### 5.2. YOLOv3-MobileNetv2 Assessment

The improved SPP unit based on YOLOv3-MobileNetv2 (SPP-YOLOv3-MN) and the improved SPP unit and GIoU loss based on YOLOv3-MobileNetv2 (SPP-GIoU-YOLOv3-MN) were compared with YOLOv3-MobileNetv2.

Figure 12 shows the training and validation losses for YOLOv3-MobileNetv2 and SPP-YOLOv3-MN (SPP-GIoU-YOLOv3-MN was trained on the basis of SPP-YOLOv3-MN, such that it was not added to the comparison here). SPP-YOLOv3-MN converged slightly faster than YOLOv3-MobileNetv2 but both the training and validation losses for the former were much smaller than that for the latter. Additionally, the training process for the former was more stable and the decline in loss was relatively smooth. Figure 13 shows the PR curves for YOLOv3-MobileNetv2, SPP-YOLOv3-MN, and SPP-GIoU-YOLOv3-MN. The area under the PR curve for SPP-GIoU-YOLOv3-MN was slightly larger than that for SPP-YOLOv3-MN and much larger than that for YOLOv3-MobileNetv2.

Compared with the YOLOv3-MobileNetv2 model, adding an SPP unit at the end of the first predicting stage resulted in an improvement of 0.92% for the absolute AP, with an increase of only 21.7% for the weight parameter and a decrease of 0.2 FPS in the detection speed (Table 7). More importantly, when GIoU was used instead of the original MSE to compute the location loss, the model achieved an improvement of 0.70% absolute AP with no parameter increase or speed reduction. By adding an SPP unit and replacing the MSE loss with the GIoU loss, SPP-GIoU-YOLOv3-MN achieved an improvement of 1.62% absolute AP, with an increase of only 21.7% in the model parameters and a negligible reduction in speed.

Table 8 compares SPP-GIoU-YOLOv3-MN with the YOLOv3 model based on ResNet50 (YOLOv3-ResNet). SPP-GIoU-YOLOv3-MN had an AP 0.80% higher than YOLOv3-ResNet. Furthermore, its model parameters were much smaller, the detection speed was 7.1 FPS faster, and the *F2* score was 0.67% higher. Overall, the former was slightly more accurate and much faster than the latter. Figure 14 shows the partial detection results for SPP-GIoU-YOLOv3-MN using the testing dataset.

### 5.3. SPP-GIoU-YOLOv3-MN Model Performance with Complete UAV Images

Complete UAV images contain extensive and complex backgrounds, which are quite different than the 416 × 416 pixels photos contained in the test dataset used herein. Detection was run on 50 complete UAV images (5472 × 3648 pixels containing 0, 1, or more poppy plots) using the SPP-GIoU-YOLOv3-MN model. This took ~110 s for completion, which is much faster than manual visual interpretation (~1000 s). However, the detection results for complete UAV images were of lower quality than that for the test dataset, with a false detection rate of 0.28 and a missed detection rate of 0.15 (Figure 15).

## 6. Discussion

### 6.1. Testing One vs. Three SPP Units

In the tests described above, an SPP unit was only added in the first prediction stage (Figure 8). However, as there were three different prediction stages, it was unclear whether adding three different SPP units of various filter sizes for the three stages would produce better results. Therefore, a new three-unit SPP3-YOLOv3-MN model was tested against the single-unit SPP-YOLOv3-MN model using the UAV poppy dataset (Figure 16).

Figure 17 shows the training and validation losses for the two models. The training process for SPP3-YOLOv3-MN was similar to that for SPP-YOLOv3-MN but there was more rapid convergence. However, both the training and validation losses for the latter model were smaller than those for the former. Determining which model had a larger area under the PR curve (Figure 18) proved difficult however, at a higher confidence the former model was more accurate, whereas the opposite was true at a lower confidence. The average precision of the former model was only 0.16% lower than the latter. Considering the uncertainty in the training, it is suggested that the precision was equal in practical terms. However, the model parameters for the former were slightly larger and the speed was slower (Table 9).

The filter sizes for the SPP unit in SPP3-YOLOv3-MN were 13 × 13, 7 × 7, and 5 × 5; 26 × 26, 13 × 13, and 9 × 9; and 52 × 52, 26 × 26, and 18 × 18 (Section 3.2.1). However, as the feature map sizes in the three stages were 13 × 13, 26 × 26, and 52 × 52, the filter sizes of the SPP unit in the second and third stages were similar to the feature map sizes in the first and second stages. This indicated that the features extracted by the SPP unit in the second and third stages were more similar to the former whereas the accuracy was more similar to SPP-YOLOv3-MN. Additionally, in the second and third prediction stages, up-sampling could lead to high-frequency information degradation and missing edge information, such that adding SPP units could only extract certain repeated texture information. In summary, adding two SPP units resulted in speed loss without a significant increase in AP.

### 6.2. Limitations of the Current Training Dataset

#### 6.2.1. Poppy Complexity

Growth stage and altitude affect poppy identification, i.e., both lower altitudes and detection during the flowering stage ease the identification process. Directly labeling all samples containing opium poppy as poppy may affect the neural network’s ability to learn characteristics and thus affect model performance for actual UAV images. Therefore, the performance of the SPP-GIoU-YOLOv3-MN model with actual UAV images may be improved by instead labeling images by growth stage and flying height.

#### 6.2.2. Background Complexity

The scenes captured by real UAV images are much more complicated than the cropped images of 416 × 416 pixels included in the dataset. As shown in Figure 19, the background of the UAV images can include complex objects, such as buildings, other crops, shrubs, and flowers that interfere with poppy identification. The complex background in complete UAV images caused high false detection rates. Therefore, we must add negative samples to the training dataset to adapt to the complex background environment by enlarging the dataset and reducing false positives.

### 6.3. Advantages and Applicability of the Proposed Method

The proposed technique is applicable to poppy identification at the seedling and flowering stages at flying heights < 200 m. Using MobileNetv2 as a backbone network simplifies the model and accelerates its forward propagation. The added SPP unit enhances the model’s ability to detect large targets and using GIoU to calculate the bounding box regression loss yields enhanced accuracy. In theory, these improvements are applicable well beyond the narrow scope of UAV poppy detection; they could be applied to improve identification of other targets, such as ships, buildings, or vehicles.

### 6.4. Model Acceleration and Future Work

Although the model had a fast detection speed on the test dataset (up to 29 FPS), it took approximately 2.2 s to analyze a complete UAV image (e.g., 6000 × 4000 pixels). The sliding window method retains the majority of the image’s information but significantly affects the detection speed. One method to improve this defect is to prune the model. This method, however, cannot fundamentally solve the problem because of restrictions associated with the sliding windows methods. Due to the sparsity of poppy plots in UAV images, the sliding window method involves many unnecessary operations performing detections in a large number of poppy-free windows. To fundamentally hasten model detection, the occurrence of such unnecessary operations need to be reduced. In future studies, the authors intend to (1) build a new detection framework to accelerate model detection, which will directly input complete UAV images without using the sliding window method, (2) use the CNN to extract a mask that may contain poppies, and then (3) conduct accurate detection within the mask. If there are no poppies in the initial image, the improved detection framework should skip the second phase, which would significantly reduce detection time.

## 7. Conclusions

The use of UAV systems to detect opium poppy plots has become a main approach to poppy surveillance. This method, however, currently relies mainly on manual visual interpretation of the images. Here, we developed a novel object detection network (SPP-GIoU-YOLOv3-MN) for use in poppy detection and achieved an AP of 96.37% and detection speed of 29 FPS using the test dataset. This proposed method significantly accelerates poppy detection and is applicable at the seedling and flowering stages at flying heights < 200 m. The proposed model also demonstrates an upgrade to the current YOLOv3 model for the detection of other objects in UAV or satellite remote sensing images. However, the use of sliding windows produced a large number of images without poppies, greatly limiting the model’s detection speed. In future studies, we intend to develop a two-stage network in which the first stage is used to extract the foreground and the second stage is used to accurately extract the poppy position.

## Figures and Tables

**Figure 1 sensors-19-04851-f001:**
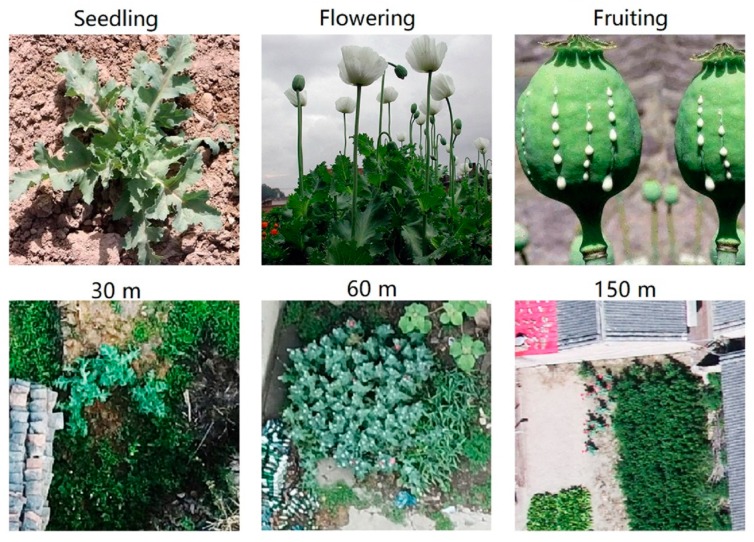
The characteristics of poppies at different growing periods and flying heights.

**Figure 2 sensors-19-04851-f002:**
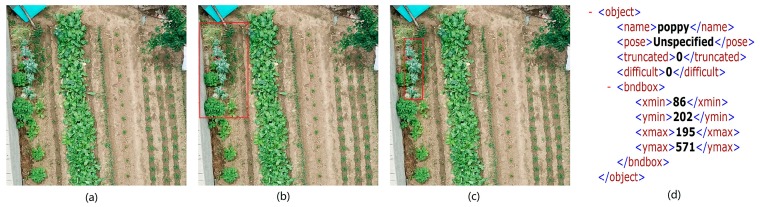
Preliminary processing for poppy selection: (**a**) original images; (**b**) verified images; (**c**) labeled images; (**d**) labeled information.

**Figure 3 sensors-19-04851-f003:**
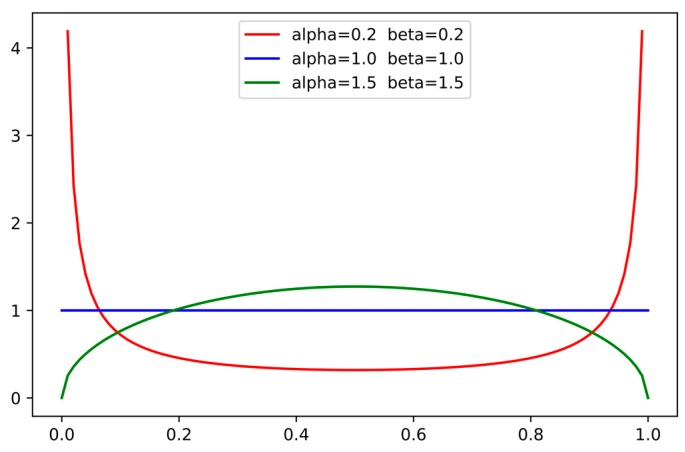
Probability density function of the Beta distribution for different values of parameters alpha and beta.

**Figure 4 sensors-19-04851-f004:**
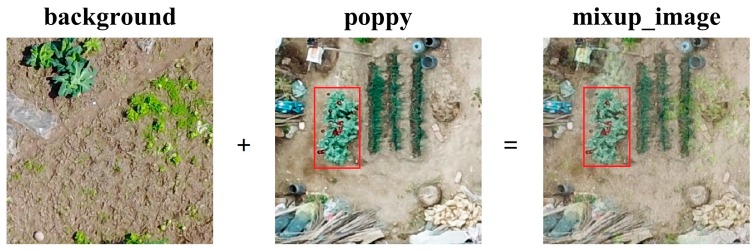
Fused image resulting from synthesizing the background and poppy images, for poppy detection based on the Beta distribution.

**Figure 5 sensors-19-04851-f005:**
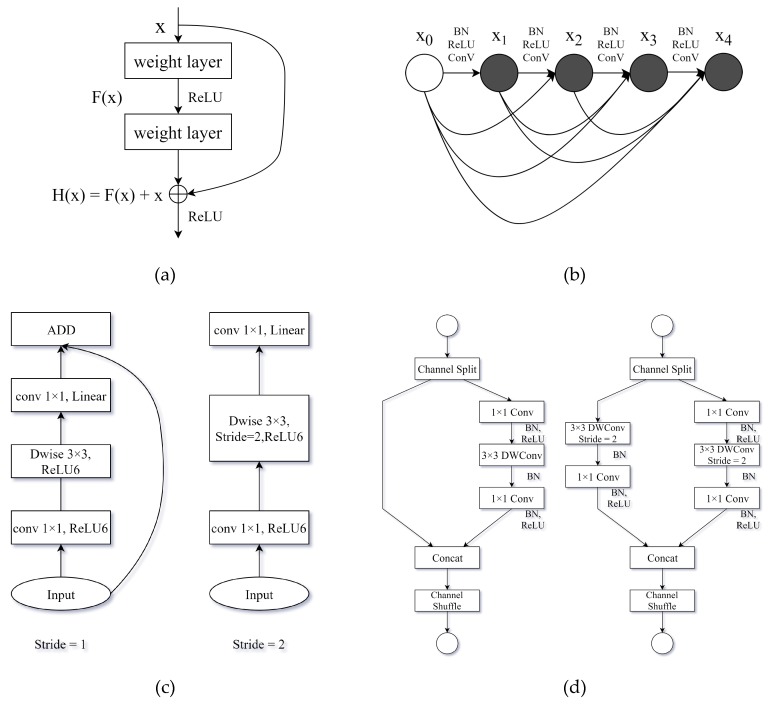
Various backbone networks: (**a**) the shortcut connection structure in ResNet; (**b**) the dense connection in DenseNet; (**c**) the inverted residual structure in MobileNetv2; (**d**) the channel shuffle structure in ShuffleNetv2.

**Figure 6 sensors-19-04851-f006:**
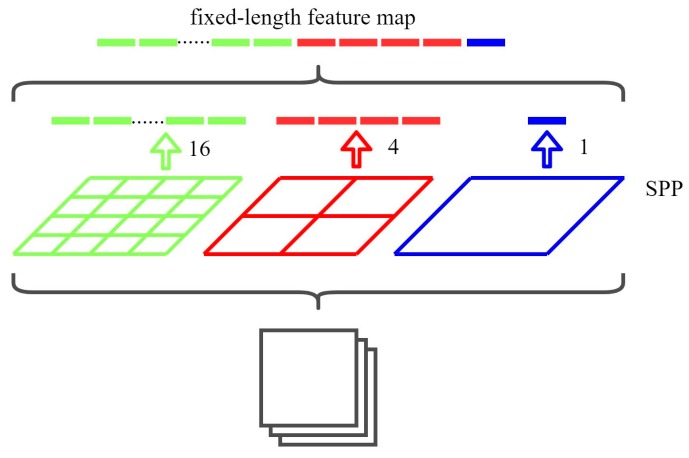
Original spatial pyramid pooling (SPP) structure in the SPP net.

**Figure 7 sensors-19-04851-f007:**
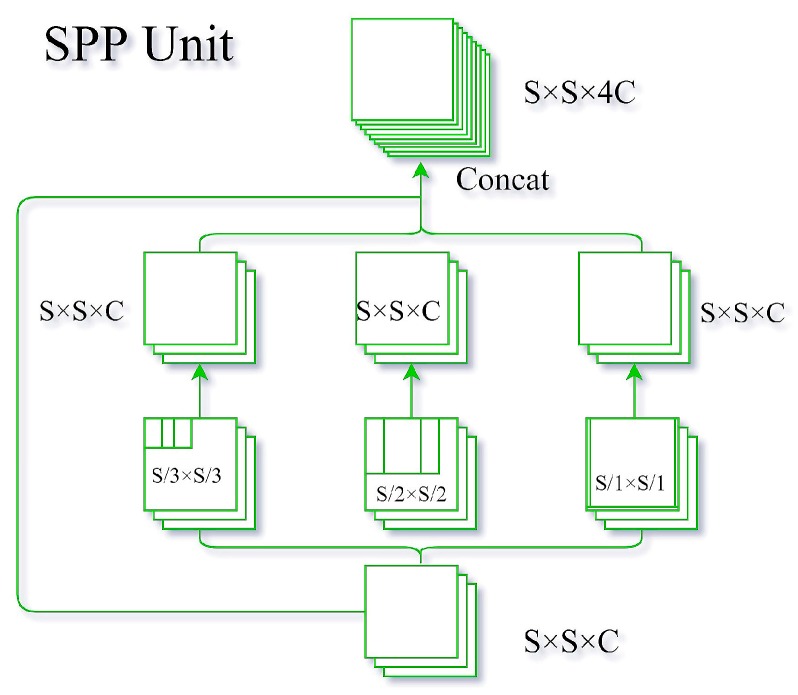
Improved SPP unit.

**Figure 8 sensors-19-04851-f008:**
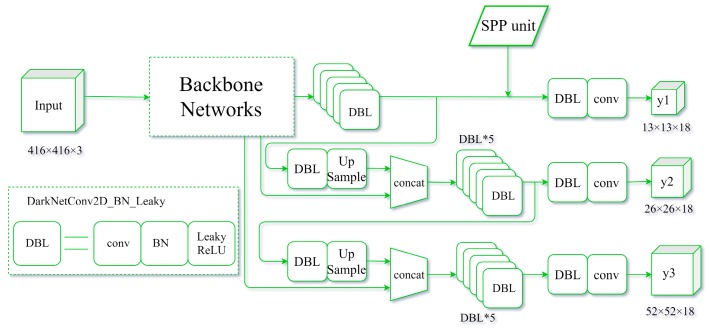
Structure of the SPP-YOLOv3 model.

**Figure 9 sensors-19-04851-f009:**
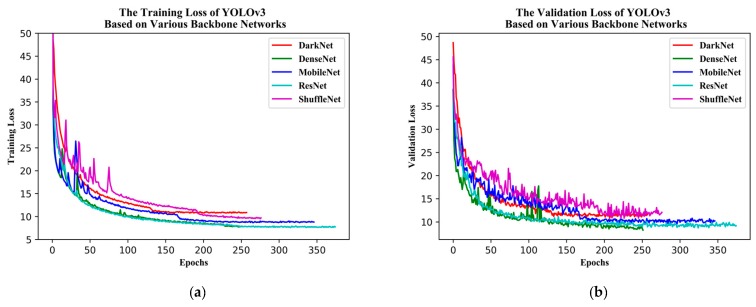
(**a**) Training and (**b**) validation loss of YOLOv3 based on various backbone networks.

**Figure 10 sensors-19-04851-f010:**
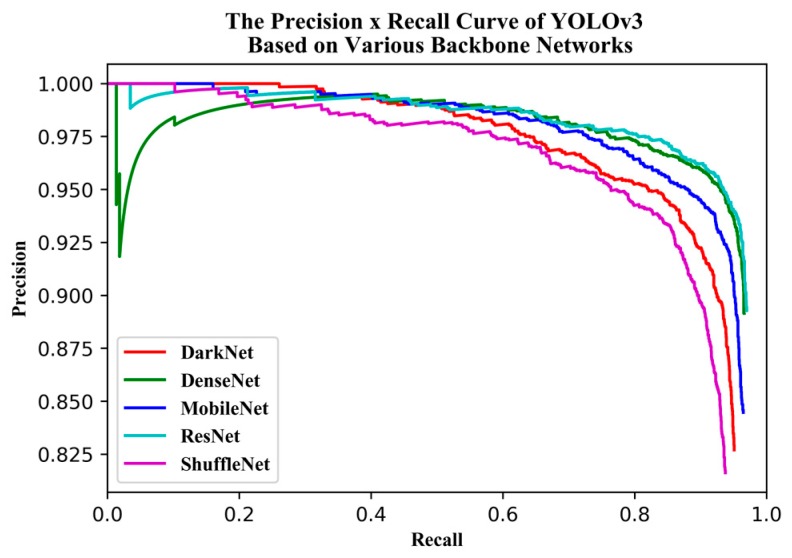
Precision × Recall (PR) curve of YOLOv3 based on various backbone networks.

**Figure 11 sensors-19-04851-f011:**
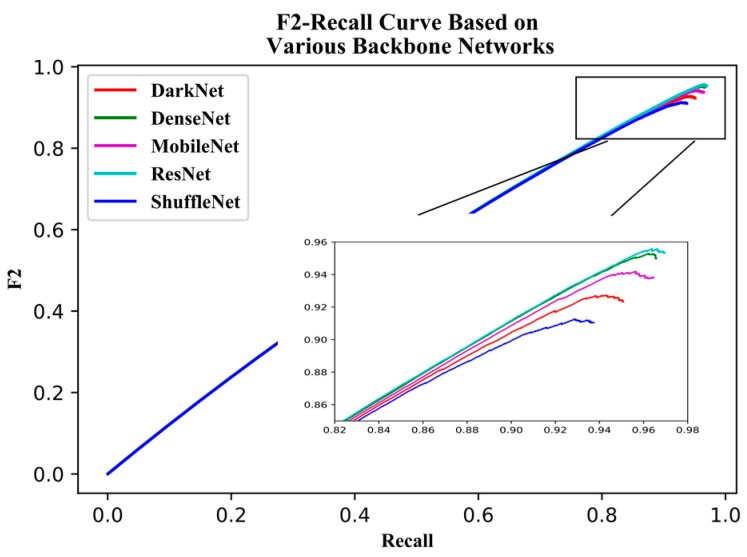
*F2* Score-Recall curve of YOLOv3 based on various backbone networks.

**Figure 12 sensors-19-04851-f012:**
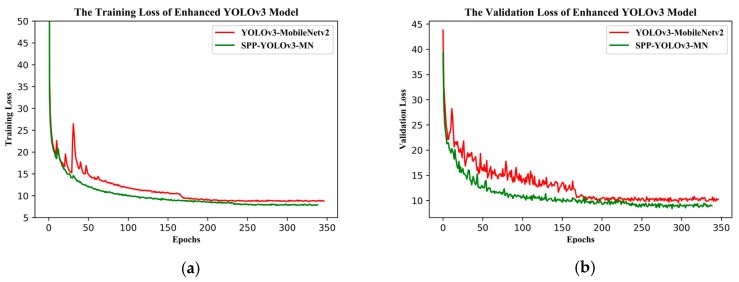
(**a**) Training and (**b**) validation losses of the enhanced YOLOv3 model.

**Figure 13 sensors-19-04851-f013:**
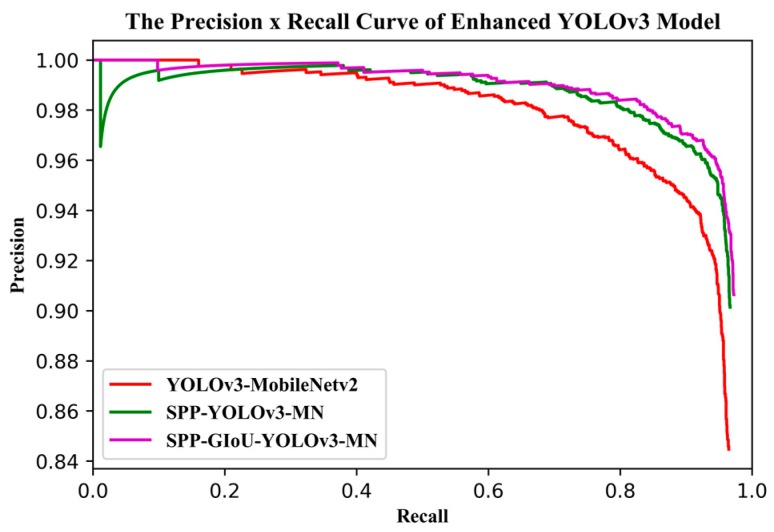
PR curve of the enhanced YOLOv3 model.

**Figure 14 sensors-19-04851-f014:**
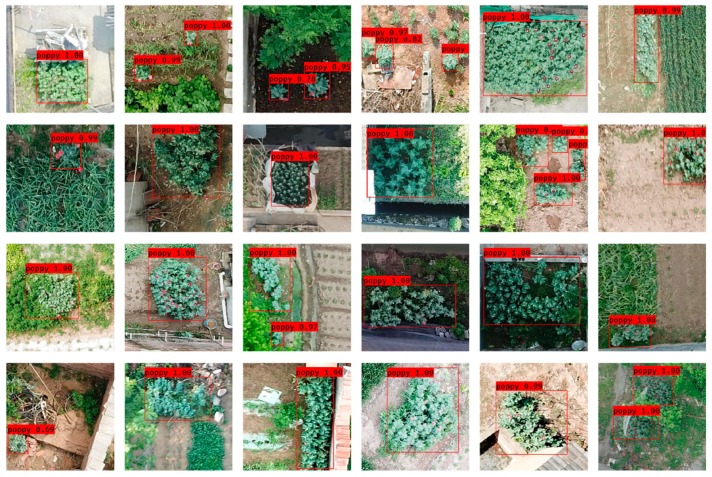
Partial detection results of the SPP-GIoU-YOLOv3-MN model using the testing dataset.

**Figure 15 sensors-19-04851-f015:**
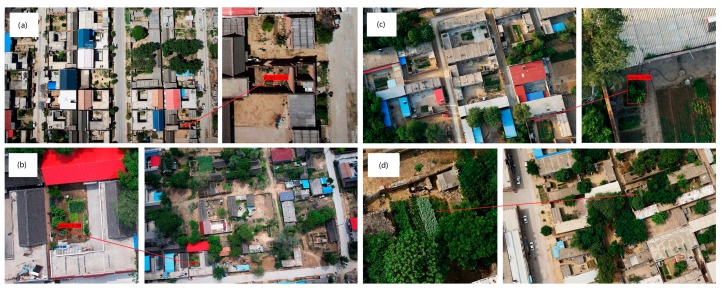
Partial test results for complete unmanned aerial vehicle (UAV) images: (**a**,**b**) the true detection; (**c**) false detection; (**d**) missed detection.

**Figure 16 sensors-19-04851-f016:**
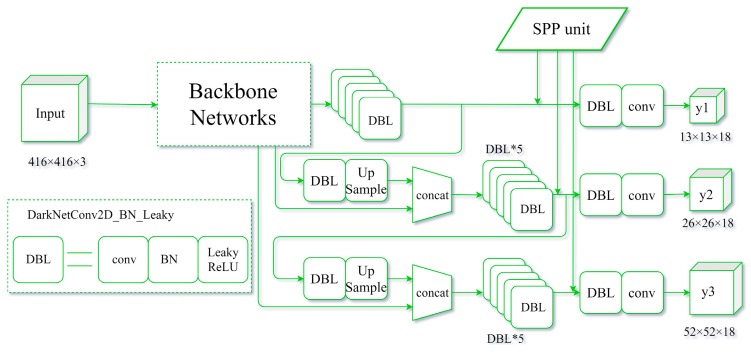
Structure of SPP3-YOLOv3-MN model.

**Figure 17 sensors-19-04851-f017:**
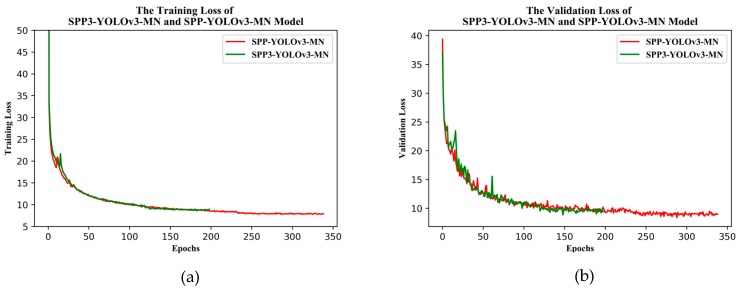
(**a**) Training and (**b**) validation losses for the SPP3-YOLOv3-MN and SPP-YOLOv3-MN models.

**Figure 18 sensors-19-04851-f018:**
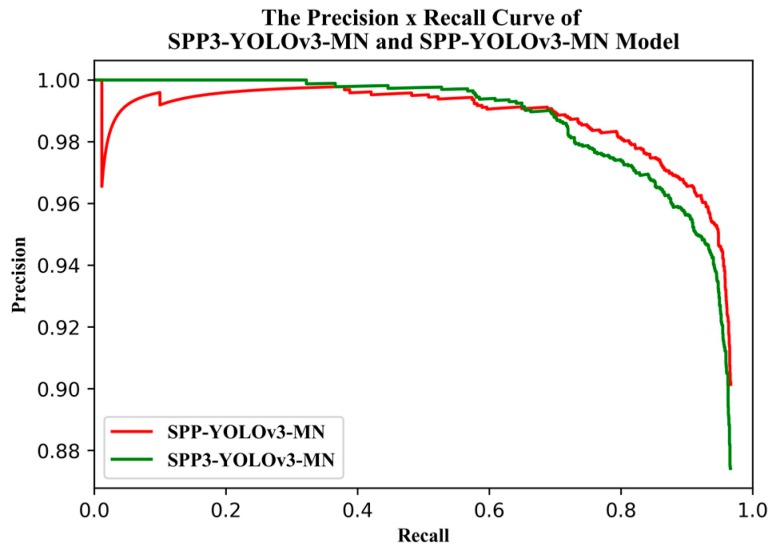
PR curve of the SPP3-YOLOv3-MN and SPP-YOLOv3-MN models.

**Figure 19 sensors-19-04851-f019:**
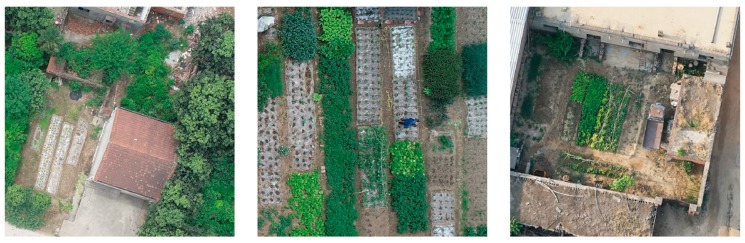
Complexity of the background in UAV images, including buildings, other crops, and shrubs.

**Table 1 sensors-19-04851-t001:** Image quantities before and after balanced.

	Flying Height	30 m	60 m	150 m
**Before Balanced**	**Seedling**	216	143	42
	**Flowering**	279	252	108
**Balanced**	**Seedling**	216	234	126
	**Flowering**	279	252	128

**Table 2 sensors-19-04851-t002:** Size of training, validation, and testing datasets.

	Training	Validation	Testing
**Number of Images**	2975	425	850

**Table 3 sensors-19-04851-t003:** The hyperparameters for the training of You Only Look Once version 3 (YOLOv3) based on various backbone networks.

Item	Value
Optimization Method	Adam
Initial Learning Rate	0.001
Learning Rate Schedule	Validation loss does not decline for 20 epochs, the learning rate increases by 0.1
Batch Size	Nearly 10 but six for ResNet and DenseNet
Training Epochs	500
Early Stopping	Validation loss does not decline for 50 epochs

**Table 4 sensors-19-04851-t004:** Other network hyperparameters for enhanced YOLOv3 model training.

Item	Value
Optimization Method	Adam
Initial Learning Rate	0.001
Learning Rate Schedule	Validation loss does not decline for 20 epochs, the learning rate increases by 0.1
Bath Size	8
Training Epochs	500
Early Stopping	Validation loss does not decline for 50 epochs

**Table 5 sensors-19-04851-t005:** Convergency epochs of YOLOv3 based on various backbone networks.

Backbone Networks	DarkNet53	DenseNet121	ResNet50	MobileNetv2	ShuffleNetv2
Convergency Epochs	257	251	374	346	276

**Table 6 sensors-19-04851-t006:** Model comparison between the YOLOv3 model based on various backbone networks.

Backbone Networks	AP ^1^ (%)	Params (MB)	Testing Time ^2^ (s)	Speed (FPS)	*F2* Score (max)
**DarkNet53**	93.00	241.1	32.7	26.0	0.927
**DenseNet121**	95.14	110.4	35.0	24.3	0.953
**ResNet50**	95.60	419.6	38.9	21.9	0.956
**MobileNetv2**	94.75	136.3	29.1	29.2	0.942
**ShuffleNetv2**	91.09	80.1	25.5	33.3	0.913

^1^ AP refers to the AP50, which indicates an average precision when the IoU threshold was set to 0.5. ^2^ The testing time refers to the time tested on the 850 testing samples.

**Table 7 sensors-19-04851-t007:** Model comparison between the enhanced YOLOv3 models and the original model.

Improvements	YOLOv3-MobileNetv2	SPP-YOLOv3-MN	SPP-GIoU-YOLOv3-MN
**SPP unit?**		√	√
**GIoU?**			√
**AP (%)**	94.75	95.67	96.37
**Params (MB)**	136.3	165.9	165.9
**Testing time (s)**	29.1	29.3	29.3
**Speed (FPS)**	29.2	29.0	29.0
***F2* score (max)**	0.942	0.955	0.960

**Table 8 sensors-19-04851-t008:** Model comparison between SPP-GIoU-YOLOv3-MN (GIoU: Generalized Intersection over Union, MN: MobileNetv2) and YOLOv3-ResNet.

Evaluation Index	SPP-GIoU-YOLOv3-MN	YOLOv3-ResNet
**AP (%)**	96.37	95.6
**Params (MB)**	165.9	419.6
**Testing time (s)**	29.3	38.9
**Speed (FPS)**	29.0	21.9
***F2* score (max)**	0.960	0.956

**Table 9 sensors-19-04851-t009:** Model comparison between SPP-YOLOv3-MN and SPP3-YOLOv3-MN.

Model	AP (%)	Params (MB)	Testing Time (s)	Speed (FPS)	*F2* Score (max)
**SPP-YOLOv3-MN**	95.67	165.9	29.3	29.0	0.960
**SPP3-YOLOv3-MN**	95.51	175.1	30.2	28.1	0.954

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
