# Peer review of "Improved UAV Opium Poppy Detection Using an Updated YOLOv3 Model"

_sensors, 2019, doi:10.3390/s19224851_

Round 1

Reviewer 1 Report

The paper has a high plagiarism rate: 50.23 % 

Please, at the end of the introduction, please insert an additional paragraph depicting the next sections.

Some figures need improved resolution and a crispy look because they appeared burred when we zoom them.

Have you thought about introducing a pre-processing stage before the backbone networks? This could reduce the complexity of the raw input images.

In your work, you state you use a detection speed of 29 FPS. Since you need to perform image fusion, why do you use such high rates? If there is a possibility of reducing this rate, then it may help relieving your computational load.

Have you thought about dividing your input image into Regions of Interest (ROIs)? Since your system does not follow people but observes a cultivation area, then you could work much better with the ROIs.

Reviewer 2 Report

This manuscript studies the detection and monitoring of poppy at a long distance, and designs a detection network based on YOLOv3 and a complete technical framework. The authors had conducted a very good preliminary research, and the technical solutions are complete and comprehensive. The author describes in detail of the task requirements, data acquisition, labeling, data enhancement, selection of backbone network, optimization of YOLOv3 network structure, design of Loss function and selection of matrics, and establishes a complete set of technical frameworks. The rationality of the scheme and the effectiveness of the algorithm are verified by experiments. The experiments are rigorous, with sufficient description of the technical details. A lot of interesting tricks are utilized to solve engineering problems. Therefore, I think this manuscript satisfies both the requirements of theoretical and applied innovation. For full acceptance, the following questions need to be resolved:

Does the method of balancing the number of samples described in lines 156-156 refer to the direct copying of certain samples? What is the theoretical basis for this? Is it necessary to experiment to justify this operation?

Is the parameter selected in the Beta distribution the optimal value? How is it determined?

I think the description of various backbone networks in section 3.1 can be shorter because they are all common networks.

In line 501, is there an "and" missing in "26*26" and "18*18"?

Reviewer 3 Report

The paper deals with an updated YOLOv3 model to detect opium poppy from UAV imagery. The topic is of interest in the scientific community and thus suitable for the Journal.

However, the paper needs major revisions before publication. The main criticalities (minors and majors) encountered in the paper are:

Please, consider using the passive form instead of we/We. Abstract: abbreviations (e.g. SPP) without explanation do not help the reader.

line 16, page 1: it’s not clear what was improved.

line 19, page 1: what do you mean with complete UAV images?

lines 56-57, page 2: too many references were reported in the brackets. A deep revision of the cited references is required; bibliographic research has to be extended to International Journals. The reference format has to be revised. Please, look at the Instructions for Authors (website). For example, the DOI number is reported twice in reference [13].

line 91, page 2: the motivation to adopt YOLOv3 has to be improved.

line 110, page 3: the pixel size was not reported. Equation 1 can be removed. Study Area and Data: no data about camera calibration, images georeferencing, coordinates of the study area, UAV speed, UAV flight time, ground markers position was reported.

lines 167-173, page 5: the definition of Beta and Alfa in not clear.

line 399, page 13: how outliers were removed? Figures 2c, 2d and 16: image quality can be improved. Figures 9-13, 17-18: legend readability is difficult (font too small). Some Figures have double grid while others do not have grid, please use the same format.

Round 2

Reviewer 1 Report

I am fine with the improvements done to the text. 

Reviewer 3 Report

Authors modified the manuscript according to the Reviewer's comments.

The quality of the manuscript has been improved.